# Optimization of Profit for Pasture-Based Beef Cattle and Sheep Farming Using Linear Programming: Model Development and Evaluation

**Addisu H. Addis** [1,2,*], **Hugh T. Blair** [1], **Paul R. Kenyon** [1], **Stephen T. Morris** [1] **and Nicola M. Schreurs** [1]

1   Animal Science, School of Agriculture and Environment, Massey University,
    Palmerston North 4442, New Zealand; h.blair@massey.ac.nz (H.T.B.);
    p.r.kenyon@massey.ac.nz (P.R.K.); s.t.morris@massey.ac.nz (S.T.M.);
    n.m.schreurs@massey.ac.nz (N.M.S.)
2   Applied Biology, College of Natural and Computational Sciences, University of Gondar, Gondar 196, Ethiopia
*   Correspondence: a.hailu@massey.ac.nz

**Abstract:** A linear programming optimization tool is useful to assist farmers with optimizing resource allocation and profitability. This study developed a linear programming profit optimization model with a silage supplement scenario. Utilizable kilograms of pasture dry matter (kg DM) of the total pasture mass was derived using minimum and maximum pasture mass available for beef cattle and sheep and herbage utilization percentage. Daily metabolizable energy (MJ ME/head) requirements for the various activities of beef cattle and sheep were estimated and then converted to kg DM/head on a bi-monthly basis. Linear programming was employed to identify the optimum carrying capacity of beef cattle and sheep, the most profitable slaughtering ages of beef cattle, the number of prime lambs (sold to meat processing plants), and sold store lambs (sold to other farmers for finishing). Gross farm revenue (GFR) and farm earnings before tax (EBT) per hectare and per stock unit, as well as total farm expenditure (TFE), were calculated and compared to the average value of Taranaki-Manawatu North Island intensive finishing sheep and beef Class 5 farming using Beef and Lamb New Zealand (B+LNZ) data. The modeled farm ran 46% more stock units (a stock unit consumed 550 kg DM/year) than the average value of Class 5 farms. At this stocking rate, 83% of the total feed supplied for each species was consumed, and pasture supplied 95% and 98% of beef cattle and sheep feed demands, respectively. More than 70% of beef cattle were finished before the second winter. This enabled the optimized system to return 53% and 188% higher GFR/ha and EBT/ha, respectively, compared to the average values for a Class 5 farm. This paper did not address risk, such as pasture growth and price fluctuations. To understand this, several additional scenarios could be examined using this model. Further studies to include alternative herbages and crops for feed supply during summer and winter are required to expand the applicability of the model for different sheep and beef cattle farm systems.

**Keywords:** linear programming; profit optimization; pasture utilization; sheep and beef farm; slaughter age





## 1. Introduction

The predominant sheep and beef cattle production system in New Zealand is pasture-based, where pasture provides up to 95% of the diet of both sheep and beef cattle [1,2], enabling New Zealand to have a low-cost and economically sustainable sheep and beef production system [2]. Sheep and cattle are complementary for pasture and management of animal health [1,3].

Pastoral sheep and beef cattle production systems are complex and are affected by many external factors [4]. It is not feasible or practical to design experiments to investigate every aspect of the potential interactions between factors [4]. Therefore, computer simulation can play an important role in gaining a better understanding of the sheep and beef

cattle production system, and the relationships within and between factors by allowing an in silico representation of the natural system [4–7]. Computer simulation can also be used to identify production constraints and enable the discovery of alternative solutions [5,7]. However, a simulation on its own does not identify the optimum production potential within given constraints [8].

Mathematical optimization models play a significant role in understanding biophysical relationships in a complex system and allow the optimization of livestock production [6,7,9–16]. These models have the capability to identify the optimum production within defined resources, to allocate resources within combinations of numerous constraints, and to suggest alternative production systems [7,17,18].

Linear programming is defined as a deterministic optimization model [9–11,17] and has been employed to optimize profitability of both dairy [7,8,10,11,16,19–21] and beef cattle farms [7,9,12,22–24]. A number of whole-farm optimization models using linear programming have been developed, including the Enviro-Economic Model (E2M) for New Zealand pasture-based dairy farms [11,18,25–27], the Model of an Integrated Dryland Agricultural System (MIDAS) and the Model of an Uncertain Dryland Agricultural System (MUDAS) in Western Australia [28–30], and the Grange Dairy Beef Systems Model (GDBSM) in Ireland and Scotland [7,31–33]. Linear programming assumes a linear association between factors; however, some of the factors in the dairy and beef cattle industries are nonlinear. Nonlinear optimization models for New Zealand pasture-based dairy farms were developed [34–37]. In terms of practical usage and identification of optimum outcomes for users, linear programming is useful [7,9,12,25,31–33,38]. With appropriate discretization of inputs, it provides reliable outputs, the ability to plan for the optimal use of resources [15,25,32,39], and discern elusive enterprise interactions often missed or poorly represented in marginal analysis [25,38,40].

An optimization tool that enables farmers to improve the profitability of sheep and beef farm within given resources would be useful to assist with improving the best allocation of resources. FARMAX (www.farmax.co.nz) is widely employed for whole-farm simulation modeling in New Zealand; however, it does not optimize [25]. Several linear [8,11,18,26,27] and nonlinear [34–37] profit optimization models for New Zealand pasture-based dairy farms have been developed. However, to date, only a few studies which were based on dry matter consumption [10], which have considered sheep production/performance only [41–43] or beef cattle production/performance only [44], or which focused on a land-based integrated grazing farm sheep and beef production system [38] have been applied for beef cattle and sheep farms in New Zealand.

Considering the above, the current study developed a whole-farm optimization model for sheep and beef cattle based on individual animal performance and metabolizable energy of the feed resource. This approach has not been undertaken previously in New Zealand for a sheep and beef farm scenario. An optimization tool using linear programming would be useful for sheep and beef cattle farmers to optimize resource utilization and farm profitability and assist with strategic planning, decision-making, and understanding their system [25,38,39]. It would also be beneficial for researchers, extension workers, and farm advisors to suggest new solutions and optimal production system for sheep and beef cattle farmers based on the existing resources and constraints [15,30,32,39].

The specific objectives of this study were to identify the maximum carrying capacity of beef cattle and sheep for a set feed resource, and to decide the profitable marketing policies and slaughtering ages of beef cattle for a Class 5 [45] pasture-based, intensive finishing sheep, and beef cattle farm on the North Island of New Zealand. This farm class is classed as fertile, partially cultivatable, and can support high stocking rates of livestock per hectare. It is mainly oriented on finishing beef cattle and sheep, i.e., cattle and sheep directly sold to meat processing plants, in comparison to the remaining seven sheep and beef cattle farm classes in New Zealand [46,47].

## 2. Materials and Methods

### *2.1. Description of North Island Intensive Finishing Sheep and Beef Cattle Farm Class 5, Taranaki-Manawatu Region*

There are eight defined sheep and beef cattle farm classes in New Zealand, which vary based on the agro-climatic zone, farming system, stocking rate, and region [46,47]. The Taranaki-Manawatu sheep and beef cattle farming region is located on the western side of the North Island of New Zealand. It is characterized by rolling hill country and has suitable soil type and climatic conditions, which is at least partially cultivatable and has the potential for high animal production (Table 1) [38,45]. The Class 5 North Island intensive finishing sheep and beef cattle farm in the Taranaki-Manawatu region was identified as a suitable farm class to implement the proposed model for the reasons of its relatively high carrying capacity per hectare (7 to 13 stock units per hectare (su/ha)) [46,47], the presence of large numbers of cattle relative to sheep (51:49 sheep:cattle su ratio) [45], and stock policies that are mainly focused on finishing animals (i.e., buying weaners at the age of 3–6 months to grow until they attain slaughter weight) for sale to meat processing plants [46,47]. Farm Class 5 is the most relevant class to base the proposed model on as it finishes a greater proportion of beef cattle than the other farm classes, which are either oriented towards sheep production, breeding, or cropping [46,47].

**Table 1.** Average production land size, total labor units, working owners, pasture fertilizer, average number of sales of beef cattle and sheep, and financial performance of Class 5 North Island intensive finishing Taranaki-Manawatu sheep and beef farm in low and high quintiles and mean of 2018 (quintile analysis ranked by earnings before interest, tax, rent, and manager wage (EBITRm)).

| Attributes | Unit | Low Quintile | High Quintile | Mean |
|---|---|---|---|---|
| Average production land | ha | 162 | 246 | 213 |
| Total labor units | No. | 1.26 | 1.31 | 1.42 |
| Working owners | No. | 0.94 | 0.94 | 0.92 |
| Pasture fertilizer | kg/ha | 297 | 270 | 260 |
| Sales total cattle | No. | 59 | 198 | 152 |
| Sales total sheep | No. | 974 | 1612 | 1483 |
| Gross farm revenue | NZ$ | 178,535 | 476,086 | 308,630 |
| Total farm expenditure | NZ$ | 188,546 | 321,281 | 241,853 |
| Farm profit before tax | NZ$ | −10,011 | 154,804 | 66,777 |

Source: [45].

A stock unit for this study was defined as the equivalent feed consumption of a 55 kg ewe weaning one 28 kg lamb, which consumes 550 kg DM/year [41,48]. In 2018, the average number of sheep and beef cattle stock units on a Class 5 farm were 1095 sheep su and 1046 cattle su, respectively [45], which equated to 10.8 stock units (su) per hectare [45]. The average size of farm Class 5 in the Taranaki-Manawatu region is 213 hectares (ha), and of that, 7 ha is used for cash crop production, such as barley and maize, grown between September and April, and 8 ha of new pasture is sown each year [45]. In this study, the land for cash crop production and new pasture grassed area was excluded from beef cattle and sheep grazing between September and April [48]; however, it was considered as grazing land between May to August.

### *2.2. Model Components and Descriptions*

#### 2.2.1. Linear Programming

A profit maximization linear programming model was developed using Microsoft Excel Solver for a one-year horizon on a bi-monthly basis [7,8,24,49,50]. R software version 3.6.0 [51] was employed to generate graphs of inputs and outputs of the model. Mathematically, linear programming can be represented as follows [17]:

$$Z = C^T \vec{X}$$

Subject to a set of constraints:

$$A\vec{X} \leq \vec{b}$$

$$\vec{X} \geq 0$$

where:

Z: objective function (profit maximization),

$C^T$: vector of profits associated with one unit of each beef cattle and sheep activity,

$\vec{X}$: vector of activities whose levels need to be solved (number of beef cattle and sheep in each class),

A: matrix of resource coefficients that are needed by each beef cattle and sheep unit activity,

$\vec{b}$: a vector of constraints.

### 2.2.2. Model System and Description

The mixed sheep and beef cattle farm model developed in this study allocated 50% of the grazing land for beef cattle activity and the remaining area for sheep production. It comprised 19 beef cattle and sheep activities (Tables S4 and S6) and 163 constraints (Tables S7 and S8 and Equations (S1A) and (S1B)). The model consisted of a beef cattle LP interface and a sheep LP interface, each of which had input variables for each activity, had objective functions, and displayed the optimum number of animals in each activity of beef cattle and sheep, with a dashboard to display the financial performance and graphs. The number of beef cattle and sheep was determined based on the carrying capacity of the total feed supply, which included the surplus pasture harvested as silage. Silage was supplied to cattle over 2 years of age and mature ewes at a maximum of 30% of the total feed intake to ensure that the allocated kg DM did not exceed the gut fill capacity (less than 3% of live body weight), which would reduce animal intake [48]. Utilized dry matter consumption was defined as the percentage of feed eaten by the beef cattle and sheep activities from the total feed supply and was estimated separately for beef cattle and sheep. A generalization of the beef cattle and sheep activities, feed demand and supply, and financial performance represented in the model is presented in Figure 1.

The model flow chart presented in Figure 1 comprises 3 modules: feed supply, feed demand, and financial. Both the quantity and quality of pasture, which are functions of the minimum and maximum pasture mass limits, plus the energy density and the pasture utilization percentage enabled the calculation of utilizable feed supply. In each bi-monthly period, the total herbage mass was assessed. Herbage above the maximum boundary when the utilizable pasture supply exceeded animal demand was conserved as silage, which then re-entered the system as supplementary feed when animal demand exceeded the pasture supply (see the left side of Figure 1). This study did not consider other external supplementary feed supplies. Per head, feed demands for maintenance, live weight gain, and pregnancy, lactation, and growing offspring in breeding ewes and hoggets (between 4 to 16 months of age, mated at 8 months of age) were calculated for each beef cattle and sheep activity in the feed demand module (see the right side of Figure 1). The total feed demand of beef cattle and sheep was always less than, or equal to, the feed supply (Equations (S1A) and (S1B)). The sheep activity was a self-replacing system (i.e., culled ewes and rams were replaced with lambs born on the farm), and thus, replacement lambs re-entered the sheep flock with the remainder being sold as either prime (sold to meat processing plant) or store lambs (sold to other farmers for finishing). Prime steer beef and bull beef from the beef cattle activity and prime and store lambs, wool, and mutton from the sheep activity were the revenue sources. Total farm expenditure for beef cattle and sheep rearing, including silage preparation costs and purchase of weaners in the beef cattle activity, was subtracted from the total farm gross revenue to estimate farm earnings before tax (middle of Figure 1).

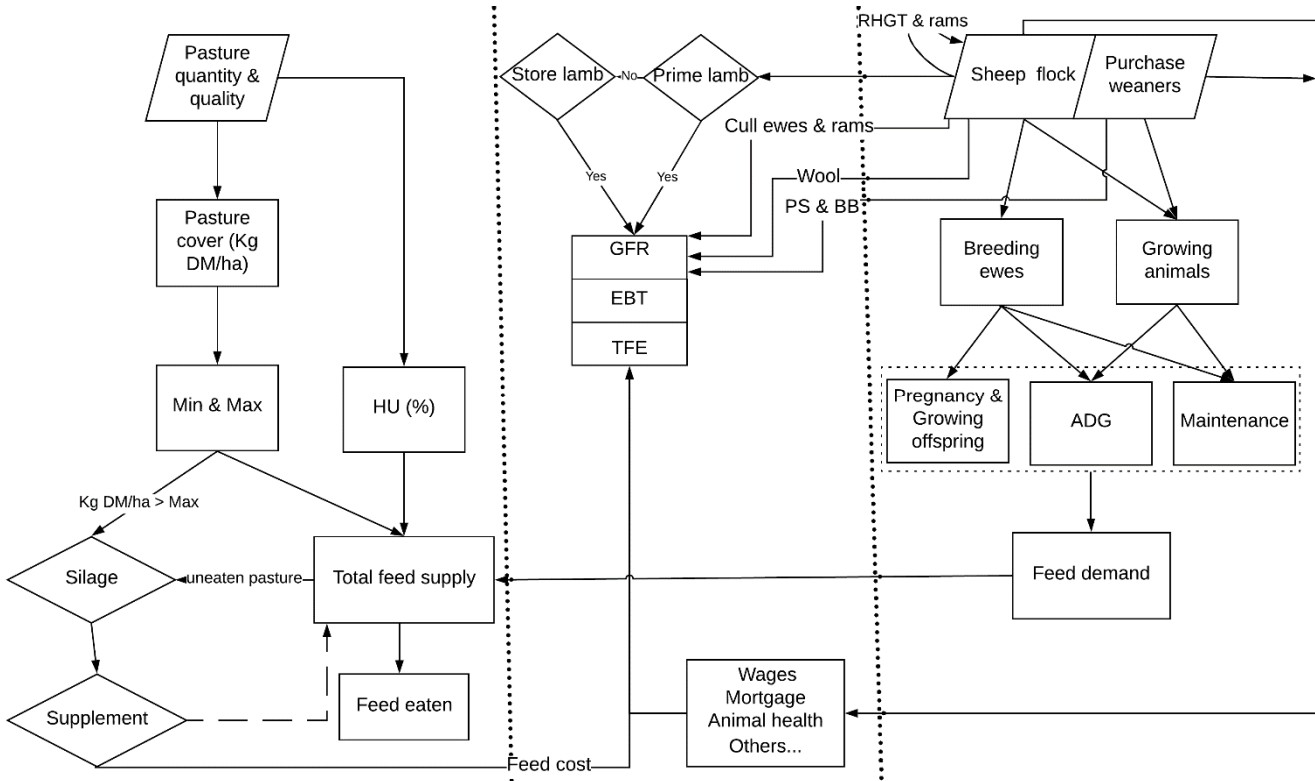

**Figure 1.** Model inputs (pasture quantity and quality parameters, beef cattle herd, and sheep flock), total utilizable feed supply, silage, and feed demand from the beef cattle and sheep activities. The outputs of the model include prime (sold to meat processing plant) and store lambs (sold to other farmers for finishing), replacement hoggets (between 4 to 16 months of age, mated at 8 months of age), cull ewes and rams, wool, prime steers (PS) and bull beef (BB), gross farm revenue (GFR), total farm expenditure (TFE), and farm earnings before tax (EBT). The rectangular boxes represent the process of the system, while activities which need decision are represented by the diamond boxes and the vertical dotted lines are drawn to separate the three main modules of the system. Min and Max: minimum and maximum pasture mass (kg DM/ha); HU: herbage utilization (%); ADG: Average daily gain; RHGT: replacement hoggets.

### 2.2.3. Inputs of the Model

#### Herbage Supply

The daily average pasture growth rates and MJ ME/kg DM for Taranaki-Manawatu by month were obtained from [47] and [52]. The same figures within a month were used in bi-monthly periods of 14, 15, and 16 days for this study (Figure 2); these were derived as 2 × 15 days for 30-day months, 15 and 16 days for 31-day months, and 2 × 14 days for February. The total average pasture mass was the sum of the post-grazing pasture mass from the previous period and the net pasture growth in a given period (Figure 3, Tables S1 and S2). The net pasture growth was calculated by multiplying average daily pasture growth rate with the number of days in the bi-monthly period (Figure 2). This study assumed the same fertilizer rate was applied as reported in [45]; this rate was not altered to manipulate the pasture growth rate. Utilizable kg DM was derived by considering the minimum and maximum pasture masses for beef cattle and sheep grazing and the percentage of pasture utilization obtained from [52] (Tables S1 and S2).

The minimum pasture mass was constrained to 1500 kg DM/ha for beef cattle grazing and 800 kg DM/ha for sheep grazing to ensure that beef cattle and sheep intakes were not restricted [38,48,53–55]. The maximum pasture masses were limited to 2500 kg DM/ha for beef cattle and 1800 kg DM/ha for sheep (Table S8, Figure 3) [38,48,55] to ensure that appropriate herbage quality was maintained and animals received the required metabolizable energy from the given kg DM of pasture [3,54]. In spring, when pasture mass for beef cattle and sheep grazing was higher than the set maximum limits and utilizable pasture mass

exceeded the animals demand, the same amount of pasture to support pasture shortage during winter was conserved as silage and was used to supplement the winter feed supply (Tables S1 and S2) [19]. The MJ ME/kg DM and utilization percentage of harvested silage were 10.5 MJ ME/kg DM and 85%, respectively [52].

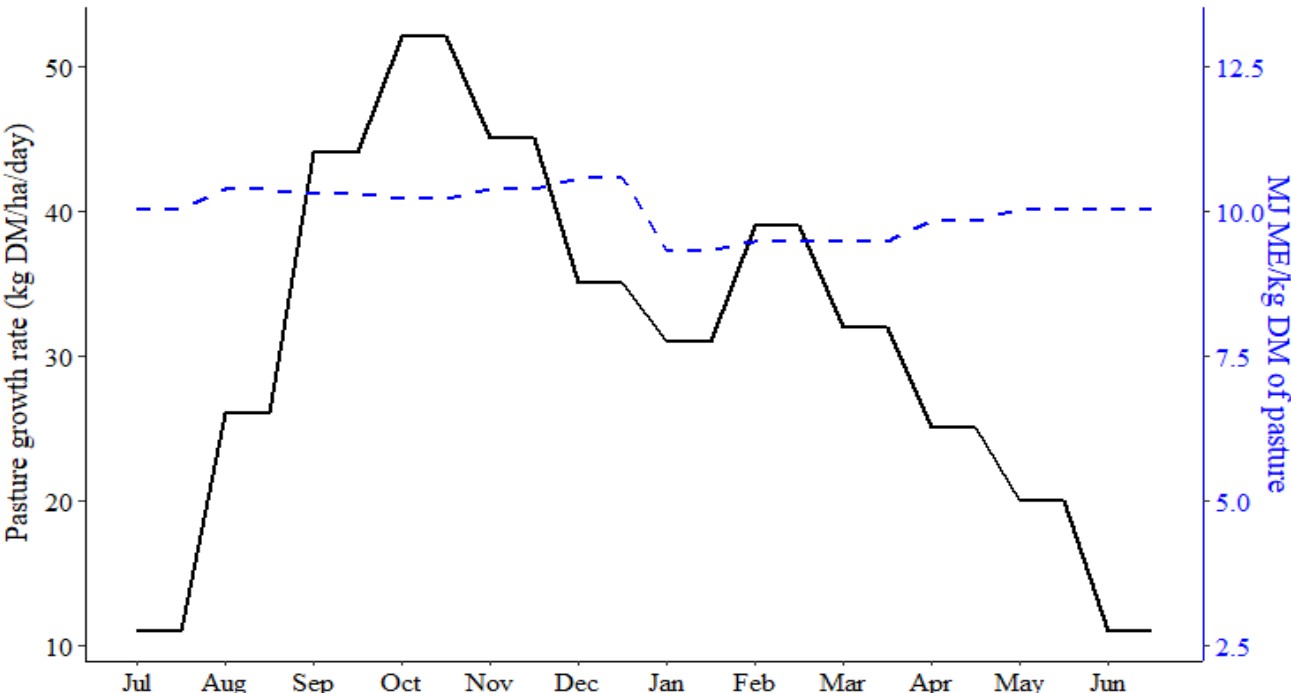

**Figure 2.** Bi-monthly average pasture growth rate kg DM/ha/day (left axis and black solid line) and MJ ME/kg DM of pasture (right axis and blue dashed line). Sources: [47,52].

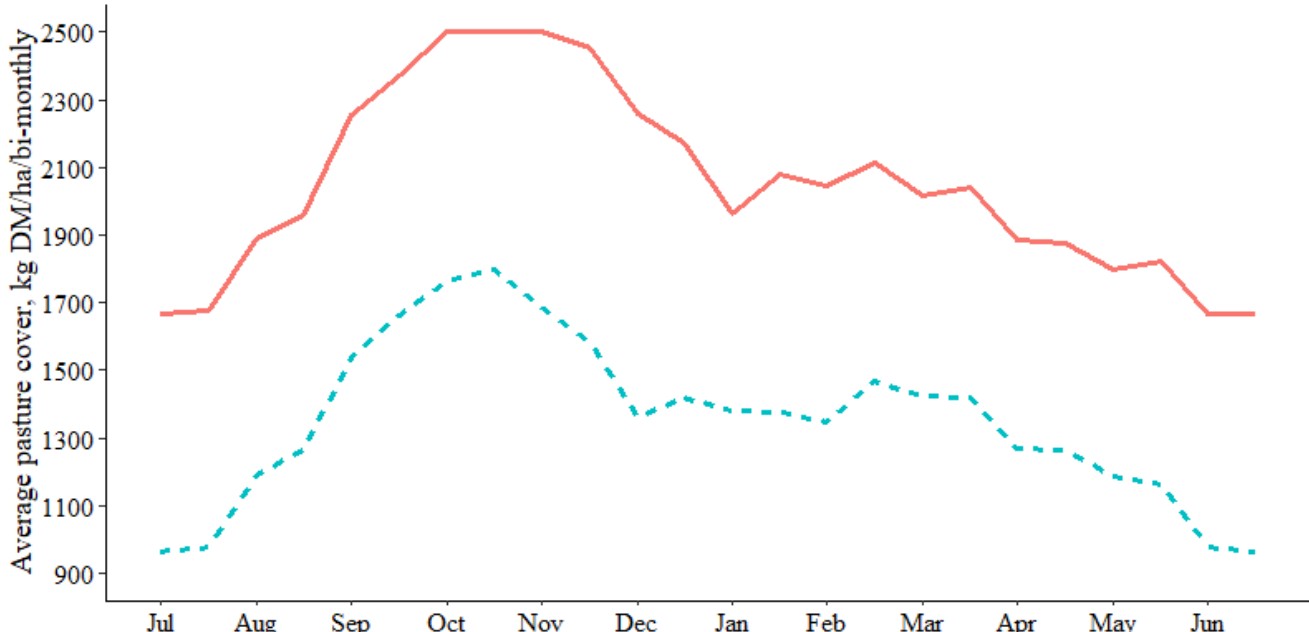

**Figure 3.** Bi-monthly average pasture mass (kg DM/ha/period) for beef cattle (orange solid line) and sheep (blue dashed line).

Beef Cattle and Sheep Activities

The beef cattle activities consisted of steer and bull finishing, which included rising-1yr (animals approaching to 1 year of age, weaners), rising-2yrs (animals aged between 1–2 years, R2), and rising-3yrs (animals aged between 2–3 years, R3) steers, and rising-1yr (weaners) and rising-2yrs (R2) bulls (Table 2) [1,45]. In New Zealand, suckling calves on beef farms, predominately Angus (47%), Hereford (14%), and their crosses (14%) [56], are weaned at six months of age. A range of beef cattle breed weaner steers at the age of six months (March), at approximately 200 kg live weight, were purchased for this study [57,58]. It was assumed that fast-growing steers attained over 500 kg live weight by 18 months, and slow-growing steers, which had 10% less average daily live weight gain than fast-growing cattle, were finished before the third winter (Figure S1).

Dairy-origin calves are weaned at three months of age [59,60]. While some dairy farmers rear and wean calves to sell them on to finishers, the majority sell their calves to calf rearers at 4–8 days of age [59,60]. Well-marked Friesian male calves are favored in New Zealand for bull beef finishing systems, as they grow faster than other breeds and classes of beef cattle [61]. Thus, this study assumed the purchase of three-month-old, uncastrated, weaner Friesian spring-born bull calves at 100 kg live weight (Table 2) [61,62].

One slaughtering option before the second winter at 18 months (S-18) and two slaughtering options at the ages of 28 and 30 months (S-28 and S-30) for steers [1,31], and four slaughtering options at ages of 16 (B-16), 18 (B-18), 20 (B-20), and 22 (B-22) months for bull were assumed (Table 2) [1,31]. Carcass weights for S-18 steers and bulls and S-28 and S-30 steers were driven using dressing out percentages of 50%, 52%, and 54%, respectively [1,31,63]. Carcasses from steers were assumed to attain the historical average price at the desired fat cover and carcass conformation classification to maximize price (NZ$5.50/CWT) and carcasses from bulls were assumed to satisfy the historical average beef price for manufacturing beef (NZ$5.25/CWT) (Table 2) [45].

**Table 2.** For steer and bull weaners, prime steer, and bull beef: the month of purchase, stock unit/head, month of sale for slaughter, age at sale, live weight (LWT), carcass weight (CWT), and unit price (NZ$/unit).

| Beef Cattle | Months of Purchase | [a] Stock Unit/Head | Month of Sale | Age at Sale | LWT (Kg) | CWT (kg) | [b] Price (NZ$/Unit) |
|---|---|---|---|---|---|---|---|
| Steer weaners | Mar. | 3.0 | - | - | 200 | - | 350.00 |
| Steer beef | | 5.0 | Feb. | 18 | 500 | 250 | 5.50 |
| | | 5.5 | Dec. | 28 | 578 | 313 | |
| | | 5.5 | Feb. | 30 | 590 | 319 | |
| Bull weaners | Nov. | 2.5 | - | - | 100 | - | 450.00 |
| Bull beef | | 5.5 | Dec. | 16 | 450 | 234 | 5.25 |
| | | 6.0 | Feb. | 18 | 498 | 259 | |
| | | 6.0 | Apr. | 20 | 531 | 276 | |
| | | 6.0 | Jun. | 22 | 550 | 286 | |

[a] equivalent stock units were estimated based on amount of feed eaten using a stock unit definition of [48]; [b] unit prices per carcass weight for sale cattle were collected from [45] and unit prices per head for weaner purchasing were collected from [64].

The sheep flock was self-replacing and included breeding ewes, rams, prime male and female lambs (i.e., sold for meat processing), store male and female lambs (i.e., sold to other farmers for finishing), and female breeding hoggets for replacement (i.e., 4 to 16 months of age), mated at 8 months of age in April to ensure new lambs in October (Table 3) [43,65]. The number of breeding ewes and rams, at a ratio of 100:1, was considered as a static population. The total number of spring-born (September) lambs was derived from the number of breeding ewes, lambing percentage (150%), and weaning percentage (80%) at a 1:1 female to male ratio [66,67]. Lambs from hoggets, which accounted for 5% of the total lambs [45], were weaned in late December. From the weaned male lambs, one-third

were slaughtered in late November, another one-third in March (prime lambs), and the rest, including hogget male lambs, were sold in late April as store lambs; the same number of culled rams was retained as replacement breeding rams (Table 3) [41–43,68]. Similarly, 40% of weaned female lambs were finished in late November (prime lambs) and the rest were either sold store in April (store lambs) or used as replacement breeding hoggets [41–43,68]. The number of replacement hoggets was assumed to match the ewe wastage (annual culls plus deaths, equating to 30% of ewes) [41,42,69]. Ewes over five years old were culled following lamb weaning in late November [41] and one-third of rams were replaced each mating season.

**Table 3.** Prime lambs (sold for meat processing), store lambs (sold to other farmers for finishing), replacement hoggets (4 to 16 months of age, mated at 8 months of age), cull ewes and rams (sold for meat processing), month of weaning and sale, stock unit/head, age at sale, and unit price ($/head).

| Sheep Classes | Month of Weaning | Month of Sale | [a] Stock Unit/Head | Age at Sale | [b] Price (NZ$/Head) |
|---|---|---|---|---|---|
| Prime lambs | Nov. | Nov. | - | 3 | 134.89 |
|  |  | Mar. | 0.4 | 6 |  |
| Store lambs | Nov.–Dec. | May | 0.5 | 10 | 97.49 |
| RHGT | Nov. |  | 1.1 | - | - |
| Cull rams |  | Apr. | 1.1 | Mixed age | 113.92 |
| Cull ewes |  | Dec. | 1.1 | Mixed age |  |
| Breeding ewes |  |  | 1.1 | Mixed age |  |

[a] equivalent stock units were estimated based on the amount of feed eaten using a stock unit definition of [48]; [b] Source: [45]. RHGT: replacement hoggets (4 to 16 months of age, mated at 8 months of age).

Estimation of Feed Demand from Beef Cattle and Sheep

Daily metabolizable energy requirements (MJ ME) for maintenance and live weight gain in growing beef cattle and sheep, as well as for pregnancy and lactation and growing offspring to weaning in breeding ewes, were estimated using equations from [52] (Tables S3 and S5). To ensure that feed demand met the pasture growth curve across seasons, the rates of the live weight gain of beef cattle and sheep were modified depending on the season [1,53,68]. The total MJ ME feed demand/head/day for each beef cattle and sheep was converted into kg DM equivalence using the energy density of the feed in each period (Figure 1) [52] and was then multiplied by the number of days to obtain the bi-monthly kg DM demand/head (Tables S4 and S6). The feed demand from beef cattle and sheep activities in terms of kg DM was constrained on a bi-monthly basis so that the total feed demand was less than or equal to the total feed supply (utilizable pasture plus silage), see Equations (S1) and (S2) [8,12,41].

2.2.4. Outputs of the Model and Evaluation

Steer and bull carcasses were the revenue sources of beef cattle activities. Mutton from culled ewes and rams (assumed to earn ewe carcass price), prime lambs which were sold to the meat processing plant at three and six months of age, store lambs (sold to other farmers for finishing), and wool production were the outputs of sheep activity (Figure 1, Table 3) [41–43]. Store lambs, prime lambs, and cull ewes and rams were assumed to earn per head the live weight schedule price (NZ$/head) (Table 3) [45]. An average of 4.17 kg of wool was produced per ewe (kg/head/year), which was valued at NZ$2.10/kg wool [45]. Linear programming was employed to identify the most profitable steer and bull slaughtering ages, prime and store lambs, and the maximum number of beef cattle and sheep that could be managed within the available feed supply.

The gross farm revenue (GFR) was defined as the total revenue earned from a one-year farm operation. It was calculated as the sum of the revenue from beef cattle and sheep activities minus the total farm expenditure (TFE) to arrive at earnings before tax per farm (EBT/farm) after which their respective per hectare and per stock unit values [43,45] were computed by dividing either by the total effective farm area (198 ha) or stock units [45].

For the purpose of developing the model, the per stock unit expenditure of various inputs from [45] were evenly distributed across the bi-monthly periods (Table S9). The total farm expenditure was computed by multiplying the per stock unit production cost with the number of beef cattle and sheep in each activity, their associated stock unit and the number of bi-monthly periods, plus baled silage processing costs (NZ$75 per large round bale) [70] (https://www.interest.co.nz accessed on 23 December 2019) and the cost of purchasing weaners for the beef cattle activity (Table S9).

There were consultation and review with a linear program expert (Barrie Ridler) Ridler et al. [11] during the model building to ensure that the model content was appropriate for the chosen farm class. Model outputs, including the pasture growth curve, feed demand and supply, and animal performance in response to the feed supply, were reviewed to ensure that they were sensible for New Zealand farming scenarios. The final outputs related to the numerical and financial performance were compared with the annual reports which were used as an evaluation benchmark [45,47].

## 3. Results

The optimized farm system carried 46% higher beef cattle and sheep stock units than that of the average value of Class 5 (Table 4). This enabled the optimized system to finish an extra 44% steers and bulls compared to that of the average value of Class 5 sheep and beef cattle farm. There were no steers slaughtered at the age of 30 months (Table 4). Similarly, a higher number of ewes (22.1%) and prime and store lambs (10.6%) were run in this study compared to the average Class 5 farm (Table 4). Unlike the average Class 5 farm, there were no store cattle and prime hoggets in the optimized system (Table 4).

**Table 4.** The number of weaner steers and bulls, stored (sold to other farmers for finishing) and finished steers (S-18, S-28, and S-30) and bulls (B-16, B-18, B-20, and B-22) (sold for meat processing), breeding ewes and rams, prime lambs (sold for meat processing) and store lambs (sold to other farmers for finishing), and prime hogget (between 4 to 16 months of age, sold for meat processing plant) and replacement hoggets (between 4 to 16 months of age, mated at 8 months of age) as well as their equivalent stock units in an average Class 5 farm and optimized system.

| Beef Cattle and Sheep and Classes | Class 5 | Optimized System |
|---|---|---|
| Steer weaners | NA | 100 |
| Stored steers | 5 | 0 |
| S-18 | 4 | 55 |
| S-28 | 43 | 45 |
| S-30 |  | 0 |
| Bull weaners | NA | 100 |
| Stored Bulls | 11 | 0 |
| B-16 |  | 7 |
| B-18 | 76 | 44 |
| B-20 |  | 36 |
| B-22 |  | 13 |
| Breeding ewes | 901 | 1100 |
| Store lambs | 251 | 345 |
| Prime lambs | 697 | 704 |
| Prime hoggets | 377 | 0 |
| Replacement hoggets | NA | 330 |
| Rams | NA | 11 |
| * Stock units | 2142 | 3141 |

* Stock unit: average throughout the year; S-18: rising-2yrs steers (slaughtered at the age of 18 months); S-28 and S-30: rising-3yrs steers (slaughtered at the ages of 28 and 30 months); B-16, B-18, B-20, and B-22: rising-2yrs bulls (slaughtered at the ages of 16, 18, 20, and 22 months); NA: no data reported.

Figure 4 shows the feed demand and supply for beef cattle activity. The annual feed demand by the beef cattle activity was supplied predominantly by pasture (95%) and only 5% by silage. Of the total feed supply allocated to beef cattle activity, they consumed 83%.

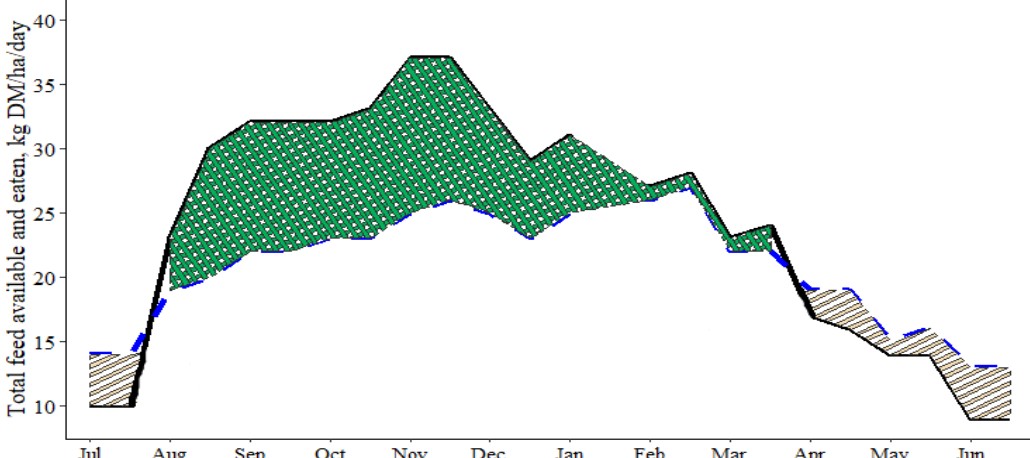

**Figure 4.** Utilizable pasture supplied (black line) and eaten by the beef cattle activity (blue dashed line which included silage) in the optimized farm system throughout the year. The excess available herbage (i.e., neither utilized nor processed for silage) is indicated by the green shaded area and the deficient available herbage (i.e., where cattle requirements were greater than the available pasture) which was supplemented with silage is indicated by the striped area.

The annual feed demand and supply for sheep activity of the optimized farm system is presented in Figure 5. Feed was predominantly supplied by pasture (98%) and the rest, 2%, was supplied by silage. Of the total feed supplied for sheep activity, 83% was consumed.

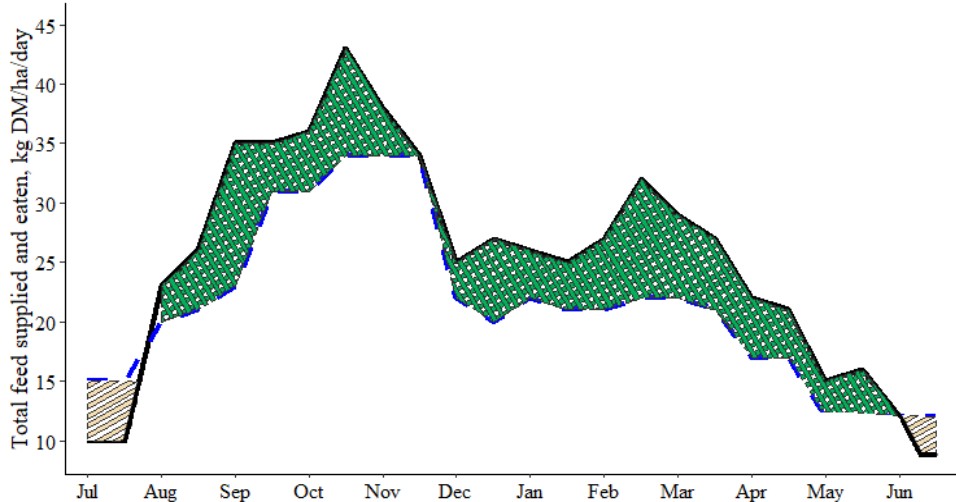

**Figure 5.** Utilizable pasture supplied (black line) and eaten by the sheep activity (blue dashed line, which includes silage) in the optimized farm system throughout the year. The excess available herbage (i.e., neither utilized nor processed for silage) is indicated by the green shaded area and the deficient available herbage (i.e., where sheep requirements were greater than the available pasture) which was supplemented with silage is indicated by the striped area.

The optimized farm system returned 53% and 188% higher GFR and EBT per hectare than that of the average values of a Class 5 farm. Total farm expenditure was 14% higher than the industry value (Table 5).

**Table 5.** Total, per hectare, and per stock unit values of the gross farm revenue (GFR), total farm expenditure (TFE), and farm earnings before tax (EBT) of beef cattle and sheep activity for an average Class 5 farm and the optimized system.

| Attributes | Unit | Class 5 | | | Optimized System | | |
|---|---|---|---|---|---|---|---|
| | | GFR | TFE | EBT | GFR | TFE | EBT |
| Beef cattle | NZ$ | - | - | - | 297,700.39 | 207,523.49 | 90,176.90 |
| Sheep | NZ$ | - | - | - | 175,820.19 | 73,789.22 | 102,030.97 |
| Total | NZ$ | 308,630.00 | 241,853.00 | 66,777.00 | 473,520.57 | 281,312.71 | 192,207.86 |
| Per hectare | NZ$/ha | 1555.20 | 1218.71 | 336.49 | 2391.52 | 1420.77 | 970.75 |
| Per stock unit | NZ$/SU | 144.11 | 112.93 | 31.18 | 150.78 | 89.57 | 61.20 |

## 4. Discussion

Linear programming can be applied to optimize beef farm profitability [3,7,9,12,22–24,38,39,50] and dairy farm profitability [7,8,10,11,19]. Grazing System Ltd. (GSL) [25], developed by [11] and later modified to become the Enviro-Economic Model (E2M), is a linear programming model, and has been used to optimize the efficiency of pasture-based dairy farm systems in New Zealand [11,18,25–27]. The two whole-farm optimization models in Western Australia, i.e., the Model of an Integrated Dryland Agricultural System (MIDAS) and the Model of an Uncertain Dryland Agricultural System (MUDAS), employ linear programing [28–30,39,40]. The Grange Dairy Beef Systems Model (GDBSM), a linear programming model, is efficient to investigate Irish beef production systems [7,31,32]. This was modified by [33] to optimize Scottish beef production systems. These applications indicate that linear programming can be a helpful tool to optimize a range of livestock production systems. This study built a profit maximization farm model using linear programming and identified the stocking rate, marketing policy, and slaughtering age of steers and bulls for feed supplied on a Class 5 North Island intensive sheep and beef cattle finishing farm in the Taranaki-Manawatu region of New Zealand.

On a mixed sheep and beef cattle farm, the two species graze separately with the aim to maximize animal performance, with each species requiring different optimum pasture masses. The sheep and beef cattle farm model developed in this study allocated 50% of the grazing land for beef cattle and 50% for sheep activity. It was assumed that the two species grazed separately, but over the year, it was assumed they grazed over all areas. The profitability of the sheep enterprise was quantified by [41,43] using 40% of the farm feed supply for cattle activity in a mixed sheep and beef cattle farm. Another study by [71] reported that approximately 40% of feed on mixed sheep and beef cattle farms should be allocated to beef cattle to ensure healthy complementarity of pasture management. The higher ratio of cattle relative to sheep in this study is due to the farm class type, with Class 5 farms having slightly more cattle relative to sheep (51:49 sheep:cattle) [45–47]. This enables Class 5 farms to return higher GFR/ha and EBT/ha compared to hard hill and hill country sheep and beef cattle farms (Classes 3 and 4) of North Island, New Zealand [45,47].

The optimized farm finished almost twice the number of steers and 32% more bulls than an average Class 5 farm. However, when compared to the high quintile values of Class 5 farms, steer and bull numbers were, respectively, 34% and nearly three-fold lower [45]. This suggests that Class 5 farmers should increase steer numbers, but decrease bull numbers, to optimize spring pasture utilization and profitability. Similarly, the number of ewes and lambs sold prime or store were 11% and 22% higher, respectively, than that of the average values of Class 5 farms [45]. These values were about 40% and 10% lower than the highest quintile values of Class 5 farms reported by the [45]. These differences are likely due to this study having optimized sheep activity based on having a self-replacing flock, and hence, all lambs were farm-born. In contrast, North Island intensive finishing sheep and beef cattle farms typically buy replacement stock to finish for slaughter [46,47]. The higher number of animals in the current model compared to the average values of Class 5 farms are likely due to the higher numbers of animals needed to optimize the carrying capacity, given the defined feed resource to ensure high herbage utilization rather

than it being left ungrazed, which would also reduce feed quality. This could also be due to some Class 5 farmers having a beef cow herd; in this case, the average reported values are not representative of a finishing farm. Labor availability, fertilizer application, land productivity differences, and management variations are likely causes of the numerical and financial differences between the optimized system and Class 5 farm statistics.

The feed demand of beef cattle and sheep does not always fit the seasonal pasture growth curve in New Zealand [44,71]. Pasture supply is high in spring and low in winter, which is a bottleneck for pasture-based sheep and beef cattle farms [44,72]. Supplementing the winter feed supply with either crops or hay/silage to carry higher stocking rates through winter is a common practice on New Zealand sheep and beef cattle farms [44,48,52]. In this study, 5% and 2% of the total feed demand for beef cattle and sheep, respectively, were supplied by silage. This enabled the modeled farm to run more livestock in winter, which improved pasture usage and reduced herbage wastage [34].

The conservation of excess pasture as silage is not used on all farms, and therefore, farmers may employ other options to control excess spring pasture. Pasture not consumed by the beef cattle or sheep nor prepared as silage in spring could alternatively be managed by increasing beef cattle and sheep number in the spring season [73] and then these animals can be sold as store or progressively finished as feed supply declines. Options include brought-in yearling or older cattle for finishing before summer [74]. Other alternatives would be to increase sheep reproductive performance, for example hogget breeding [65], buying supplementary winter feed, or growing winter forage crops for animals to allow for greater livestock numbers in winter [44]. These options were not evaluated in the current study; however, the model would be able to test the profitability of these alternative strategies if needed.

Heavier cattle requires greater feed for maintenance and live weight gain [48,52], and thus, the model finished more than 70% of beef cattle before their second winter (rising-2yrs steers and bulls) and the remaining before their third winter. Similarly, 67% of lambs were sold prime, and the rest were sold store. This enabled the modeled Class 5 farm to earn 53% and 188% higher GFR/ha and EBT/ha, respectively, than the average value of the industry data [45]. These values were 7% and 25% higher, respectively, compared to the industry high quintile of Class 5 farms [45]. The finding of [62] supported that finishing of fast-growing and younger cattle is more profitable than slow-growing, older cattle. This suggests that farm Class 5 farmers would benefit from finishing a greater number of beef cattle before their second winter, which would require farmers to consider fast-growing cattle breeds or to provide alternative feeds to grow cattle faster, to improve their farm profitability.

The model developed in this study was deterministic and considered linear relationships between feed supply and demand from beef cattle and sheep activities [7,9,12], although it is acknowledged that pasture supply is nonlinear across the year. This is a known limitation of these types of models [12]. To minimize the effect of this limitation, average live weight gains of growing livestock were adjusted to fit the feed supply curve and the feed supply and demand data were discretized into bi-monthly periods [3,12]. This allowed the model to enumerate the number of cattle and to make decisions on the optimum number of animals based on the available feed supply on a bi-monthly basis. It is important to realize that the production profile and financial performance of the model can be affected by slight changes in the bi-monthly feed supply and unit prices of beef cattle and sheep. The current model did not investigate risk and uncertainty, which could be assessed through the running of various scenarios using the model, and thus, comparing the outputs. Based on this, farmers could determine the likelihood of positive or negative outcomes [25,39].

In the current system, the potential benefit of older cattle such as cows for controlling the pasture quality for both sheep and young cattle [3] was not considered. Having a herd of beef breeding cows could improve farm profitability, as they can utilize low-quality pasture in winter. This would further reduce pasture wastage, but would also maintain

higher quality pasture to support higher growth rates in younger cattle and lambs [3]. The impacts of herbage quality, such as protein content and the use of crops to alleviate feed shortages in summer and winter seasons, were not evaluated. Hence, further studies would be required to improve this model applicability under different classes of sheep and beef cattle farms. The current profit optimization model could be extended to multi-objective linear programming to optimize land utilization, sheep:beef cattle ratio, and stocking rate for different scenarios [13,14]. This would enable users to include additional constraints and objective functions which were not considered in this study.

## 5. Conclusions

The optimized Class 5 farm model developed using linear programming identified the optimum number of beef cattle and sheep that could be managed within given feed resources for Class 5 New Zealand North Island intensive finishing sheep and beef cattle farms. The model could be employed by the farmers to understand pasture utilization throughout the year and decision making for conserving excess pasture to support winter feed supply, thereby reducing pasture wastage and improving overall pasture utilization.

The modeled Class 5 farm had nearly the same numbers of beef cattle and sheep as the industry high quantile farms. The majority of beef cattle were finished before their second winter and the majority of lambs were sold prime. This enabled the modeled farm to earn comparable GFR and EBT to the top 20% of farms.

The combined outputs suggest that the model accurately represented real farm systems, and therefore, it would be suitable for use by beef cattle finishing farmers and could be used to model other potential production systems on this class of land. The current model could be also adapted to other farm classes in New Zealand by providing the appropriate input parameters of those systems. Further study to incorporate multiple objectives, such as the stocking rate and sheep:beef cattle stock ratio, would improve the model applicability.

**Supplementary Materials:** The following are available online at https://www.mdpi.com/article/10.3390/agriculture11060524/s1, Figure S1: Live weight of bulls (dashed blue line), fast-growing steers (green round dot) and slow-growing steers (solid black line), Table S1: Land size (ha), number of days, pasture growth rate/ha/day, post-grazing and pre-grazing pasture masses (kg DM), percent of pasture utilization (%), utilizable kg dry matter of pasture (kg DM), metabolizable energy density per kg DM (MJ ME/kg DM), excess pasture for silage preparation (Silage), silage utilization percent (%) and MJ ME/kg DM of silage throughout the year for beef cattle activity, Table S2: Land size (ha), number of days, pasture growth rate/ha/day, post-grazing and pre-grazing pasture masses (kg DM), percent of pasture utilization (%), utilizable kg dry matter of pasture (kg DM), metabolizable energy density per kg DM (MJ ME/kg DM), excess pasture for silage preparation (prepared silage), silage utilization percent (%) and MJ ME/kg DM of silage throughout the year for sheep activity, Table S3: Daily per head feed demand for various classes of steers and beef bulls (kg DM/head/day), Table S4: Bi-monthly per head feed demand for different slaughtered age steers and beef bulls (kg DM/head/bi-monthly), Table S5: Per head feed demand on a daily basis for sheep flock (kg DM/head/day), Table S6: Per head feed demand on bi-monthly basis for sheep flock (kg DM/ha/bi-monthly), Table S7: Bi-monthly constraints of beef cattle and sheep numbers, Table S8: The bi-monthly minimum and maximum pasture mass constraints, Table S9: Costs of buying weaners for the beef cattle activity and the bi-monthly costs of various inputs per stock unit for beef cattle and sheep activities, Equation S1A: Total feed demand and feed supply constraints for beef cattle activity, Equation S1B: Total feed demand and feed supply constraints for sheep activity.

**Author Contributions:** A.H.A., H.T.B., S.T.M., P.R.K., and N.M.S. contributed to the conceptualization, methodology, software, validation, formal analysis, investigation, data curation, original draft preparation, and review and editing. S.T.M., H.T.B., N.M.S., and P.R.K. contributed to the supervision, project administration, and funding acquisition. All authors have read and agreed to the published version of the manuscript.

**Funding:** This research was funded by Massey University (Massey Foundation), and Addisu Addis was sponsored by a Massey University doctoral scholarship.

**Acknowledgments:** We would like to thank Massey University for financial support. Our acknowledgment is also extended to Barrie Ridler for his unreserved effort during the model development and being an external reviewer of the model during the development and while writing the manuscript.

**Conflicts of Interest:** The authors declare no conflict of interest.

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
