# Peer review of "Optimization of Profit for Pasture-Based Beef Cattle and Sheep Farming Using Linear Programming: Model Development and Evaluation"

_agriculture, doi:10.3390/agriculture11060524_

Round 1

Reviewer 1 Report

This manuscript is badly formatted and not according to the instructions to authors.

Also, it is badly written.

It is superfluous and verbose. It is an extremely long manuscript that, as it is, will not read by anyone.

Before real evaluation, the manuscript needs to become shorter and concise. Avoid too many details, they turn readers away. Also, decrease the verbose style throughout the manus. A lot of minute data (e.g., Table 4) are redundant and should, at most, be transferred to supplementary material (if not deleted altogether).

The figures are nice and worthy and should be extended if possible. They contribute positively to the manuscript.

The discussion does not cover all point in the work and also takes into account neither other similar studies in other countries, not previous relevant works in NZ. This is bad and unfair to previous workers. These studies should be referred to the Discussion and the results of the authors need to be compared to those of previous researchers.

In all: the manuscript has flaws, which do not allow proper evaluation; these need to be rectified and the revised version needs to be reviewed from scratch. It is not possible to provide a general opinion regarding possible publication, as really the evaluation was not full, given the problems indicated above. In the revised version, please avoid NZ colloquialisms and please use English expressions that can be understood by scientists worldwide.

Reviewer 2 Report

This is a well prepared and nicely described paper on application of linear programming to pasture-livestock optimisation.

It comes up with some interesting locally relevant results through some well focused application of modelling. In the broad sense there’s nothing new about this approach or the objectives “to identify maximum carrying capacity.. for a set feed resource” etc, except perhaps some regional novelty in application.

The paper should do much more to review and justify the LP approach among other alternatives (such as non-linear approaches eg https://www.sciencedirect.com/science/article/pii/S0022030213001070;

https://www.sciencedirect.com/science/article/pii/S0308521X16304590) ,

and highlight why this method is still the most suitable for purpose compared to alternatives   https://researcharchive.lincoln.ac.nz/bitstream/handle/10182/9563/The%20Usefulness%20and%20Efficacy%20of%20Linear%20Programming%20Models...pdf?isAllowed=y&sequence=1 .

More could be made of the experience of Ridler et al as reference to these studies seems somewhat shallow.

There is also a need to do more discuss the singular objective of the optimization (profit) in terms of the possibility of multi-criteria objectives and possible additional constraints of farmers. In line with this, much more has to be done to discuss how risk is represented and may influence stocking rates etc and conclusions in a real farm setting. The word risk is not mentioned in the paper. This is remarkable and needs to be addressed, particularly given debate around the value of LP optimizations generated in the absence of risk. https://www.jstor.org/stable/1349622?seq=1

https://www.sciencedirect.com/science/article/pii/S0308521X21000032

Some mention of the constraints and considerations needed in the approach is late in the paper (eg p24) but more needs to be done upfront.

For an international readership more could be done to mention LP approaches and associated livestock modelling insights from elsewhere when introducing the pros and cons of the approach.

More attention on highlighting some of the limitations of the LP approach and other constraints to pasture/livestock modelling would improve the paper and so too would an effort to highlight what findings and insights may be of value to an international readership.

Further consideration of whether some of the livestock and market terms will be readily understood by an international audience is likely to improve the paper Abstract: a mention of risk and how it is handled would be useful in the abstract as you are providing a lot of ‘definitive’ results. L18 various Activities not activity L26 clarify what 550kg DM is (you mention both pasture and silage earlier) L70-71- I assume this statement is NZ specific, so should be clarified as such. There are of course many studies in Australian systems based on sheep DM consumption and performance. L83 A bit more information on the NZ land class system in the text would be useful and class 5 in particular. L474 – is there a strong precedent for an LP model like this being used directly by farmers? Is that realistic?

Reviewer 3 Report

Well done.  The writing quality and flow of ideas is excellent.  The application of linear programming to the problem is an excellent use of the technique.  The results of the study show that the use of your model can significantly improve the economics of grazing practice in New Zealand, as well as other parts of the world with some model calibrations for local conditions.    

Author Response

The Authors are grateful for your kind reply. We did not need to reply to you back

Round 2

Reviewer 1 Report

The manuscript has been improved vastly by the changes made by the authors.

It is now in a significantly better condition than in the previous version.

The authors have done a lot of work and have made useful changes in the revised manuscript.

Before final acceptance, it needs a significant improvement in English language to avoid colloquialisms and local expressions. All the manuscript should be written in English English language, as it submitted in an international journal.

After appropriate changes, it can be accepted.
